# Research on Trajectory Prediction of a High-Altitude Zero-Pressure Balloon System to Assist Rapid Recovery

Jiwei Tang [1], Shumin Pu [2,*], Peixi Yu [2], Weicheng Xie [3], Yunfei Li [4] and Binxing Hu [3]

1. School of Aeronautics & Astronautics, Shanghai Jiao Tong University, Shanghai 200240, China
2. Shanghai Jiao Tong University Chongqing Near Space Innovation R&D Center, Chongqing 401135, China
3. Aerospace System Engineering Shanghai, Shanghai 200240, China
4. North Institute of Nuclear Technology, Xi'an 710000, China
* Correspondence: psm446734413@mail.nwpu.edu.cn; Tel.: +86-18281029537

**Abstract:** A comprehensive simulation model is established to predict the trajectory of a high-altitude zero-pressure balloon flight system with no parachute that is required to carry the load floating at the designated altitude for several hours or less. A series of mathematical models, including thermal dynamic, atmospheric, earth, wind, geometry, and exhaust models, are developed to predict the trajectory of the balloon flight system. Based on these models, the uncertainties of the launch parameters and the corresponding flight performance are simulated. Combined with the control strategy, the entire flight trajectory is simulated and discussed in detail, including the ascending, floating, and descending phases. The results show that the vertical velocity takes on a W shape during the ascent process. Furthermore, the balloon begins to gradually descend with weakening solar radiation after noon. Moreover, the landing vertical speed of the balloon flight system can approach zero with the control strategy applied, whereas the lateral drift range is more limited relative to the uncontrolled flight mode. The results and conclusions presented herein contribute to the design and operation of a zero-pressure balloon flight system within limited airspace to improve the rapid recovery ability of the flight system.

**Keywords:** zero-pressure balloon; trajectory prediction; thermal dynamic model; control strategy; rapid recovery





## 1. Introduction

With the potential for civil and military applications, including communication, observation, and scientific research, there is growing interest in studying platforms in near space [1–3]. The high-altitude zero-pressure balloon has become an indispensable test platform in near space owing to its accessibility and affordability, attracting the attention of researchers worldwide [4–8].

In order to accomplish the mission of a balloon flight system, it is necessary to predict the system's trajectory in advance to prevent the balloon from floating out of the designated airspace with the wind. Throughout the flight mission, uncertainties with respect to the launch and environmental parameters play an important role in predicting flight performance, which influences the safety of the balloon flight. The failure of helium mass prediction can result in the balloon not reaching the designated altitude. The failure of helium temperature prediction can affect trajectory planning. The effectiveness of trajectory prediction depends on the accuracy of the relevant parameters that are necessary to optimize the balloon design and launch process.

In recent years, many investigations have been conducted on high-altitude balloon trajectory predictions. Rodger E. Farley designed a software program without considering the uncertainties during the flight process [9]. Qiumin Dai established a thermal dynamic model of a super-pressure balloon, emphasizing the importance of the film radiation property and clouds with respect to the thermal behaviors of the balloon [10]. András Sóbester

developed a trajectory prediction model incorporating a new stochastic drag model based on empirical data derived from thousands of radiosonde flights to improve prediction accuracy [11]. Yi Zhang took the influences of the initial launch conditions into consideration to simulate the ascending process of a balloon [12]. Ö. Kayhan designed a gas-compress-release system with PID control for a zero-pressure balloon system, which helps the balloon system reach its target altitude and land at a safe vertical speed [13]. Yongseon Lee studied a trajectory prediction model that considers the uncertainties during the flight process [14]. Sherif Saleh developed an ascending performance prediction model that optimizes the necessary inflating quantity of helium to improve the ascending performance [15]. R. Waghela developed a balloon trajectory control system capable of passively guiding a high-altitude balloon that uses a given sail as an actuator to control the balloon system [16]. Xiaolong Deng detailed the Google Loon super-pressure balloon system, which uses a sub-ballonet for altitude control and an electric propulsion propeller system for enhanced operational performance [17]. Previous studies have mainly focused on the uncertainties with respect to thermal environmental parameters and launch parameters, which influence the accuracy of trajectory prediction. Furthermore, the previously proposed control strategies have mainly focused on altitude control.

Based on previous works, in this paper, we focus on trajectory prediction for a high-altitude zero-pressure balloon within limited airspace to optimize the rapid recovery of the balloon flight system. First, a comprehensive simulation model is established to predict flight performance and trajectory during the ascending, floating, and descending phases by considering various uncertainties. Second, taking control strategies into consideration improves the ability to rapidly recover the balloon flight system, preventing the balloon from flying out of the designated airspace in order to avoid possible accidents. Because the balloon is unpowered and there is no way to control its horizontal motion, it is necessary to predict its trajectory after launching from the ground.

The balloon flight system considered in this study descends without the separation of the balloon from the load, which enables the reuse of the balloon capsule, representing cost savings, compared with a system in which the balloon bursts, relying on a parachute to recover the load. Figure 1 shows the entire process of the flight mission of a balloon flight system, including the dynamic launching phase from the ground, the ascending phase, the floating phase, and the descending phase.

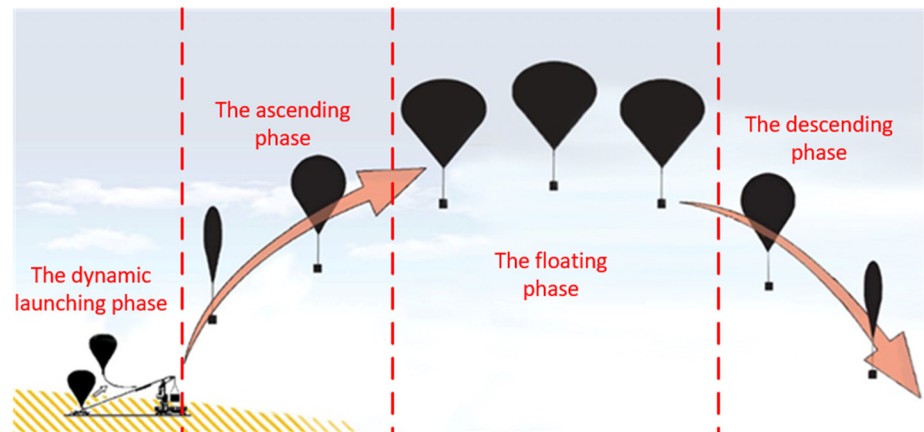

**Figure 1.** Flight mission of a balloon flight system.

## 2. Environment Model

During the flight process of a zero-pressure balloon, the differential temperature magnitude between the internal helium and the ambient atmosphere directly affects the net buoyancy of the balloon. Changes in environmental parameters, such as the surrounding atmospheric temperature, air pressure, air density, thermal radiation, and acceleration of gravity during the flight process, also directly or indirectly affect the internal helium

temperature and helium mass of the zero-pressure balloon. All of these variations should be taken into account to improve the accuracy of balloon system trajectory prediction.

### 2.1. Atmospheric Model

Atmospheric temperature, pressure, and density vary with altitude. In this study, the standard atmospheric model is used, which was released in 1976 [18]. The atmospheric temperature is defined as

$$
T_{air} = \begin{cases}
288.15 - 0.0065 * z & 0\,m < z \le 11,000\,m \\
216.65 & 11,000\,m < z \le 20,000\,m \,. \\
216.65 + 0.0010 * (z - 20,000) & 20,000\,m < z \le 32,000\,m
\end{cases}
\tag{1}
$$

The atmospheric pressure is defined as

$$
P_{air} = \begin{cases}
101,325 * ((288.15 - 0.0065 * z)/288.15)^{5.25577} & 0\,m < z \le 11,000\,m \\
22,632^{(-(z-11,000)/6341.62)} & 11,000\,m < z \le 20,000\,m \,. \\
5474.87 * ((216.65 + 0.0010 * (z - 20,000))/216.65)^{-34.163} & 20,000\,m < z \le 32,000\,m
\end{cases}
\tag{2}
$$

According to the ideal gas law, the density is defined as

$$
\rho_{air} = P_{air} / (T_{air} * R_{air}),
\tag{3}
$$

where $z$ is the altitude, and $R_{air}$ refers to the gas constant of air.

### 2.2. Thermal Model

Throughout the entire flight process of a zero-pressure balloon, the balloon system is in a thermal environment. Research [19] has shown that the main heat sources are direct solar radiation, ground-reflected radiation, sky-scattered radiation, Earth's infrared radiation, atmospheric infrared radiation, surface thermal radiation, and convective heat transfer. Figure 2 shows the heat sources for a high-altitude balloon system. Direct solar radiation, ground-reflected radiation, and sky-scattered radiation are examples of solar short-wave radiation. Earth's infrared radiation, atmospheric infrared radiation, and surface thermal radiation are examples of long-wave radiation.

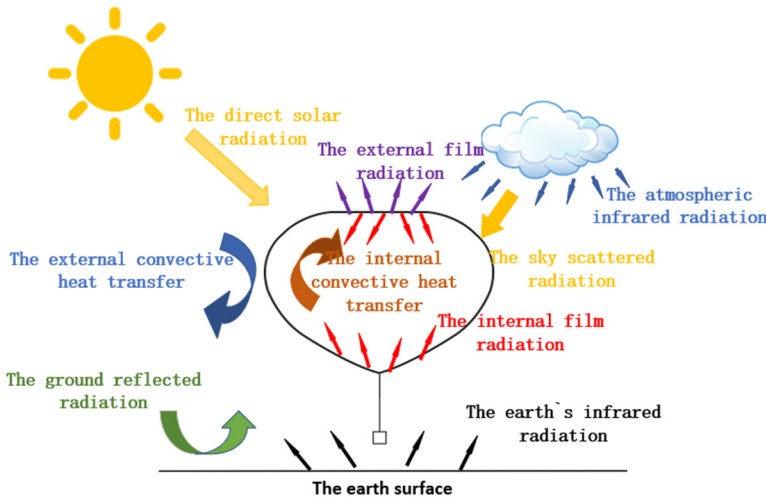

**Figure 2.** Heat sources for a high-altitude balloon system.

Through the heat transfer mechanism of the thermal model, heat is transferred among the balloon film, the internal helium, and the atmosphere [20]. Figure 3 shows the heat transfer mechanism. The following content illustrates the heat sources' models and the

heat transfer relationships among the atmosphere, the balloon film, and the internal helium in detail.

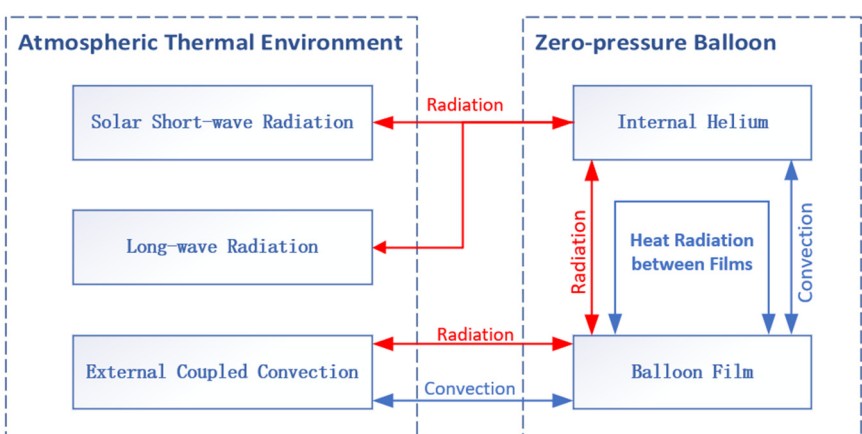

**Figure 3.** Heat transfer mechanism of the thermal model.

### 2.2.1. Direct Solar Radiation

The absorption of direct solar radiation by the balloon consists of absorption by the outer surface and absorption by the inner surface through the balloon film, and it is defined as follows [9]:

$$Q_{Direct} = \alpha * A_{projected} * I_{sun} * \tau_{atm} * \left[1 + \tau * \left(1 + r_{effective}\right)\right], \tag{4}$$

where $\alpha$ indicates the film absorption factor of solar radiation, $A_{projected}$ is the illuminated projected area, $I_{sun}$ is the solar radiation constant number, $\tau_{atm}$ is the atmospheric transmissivity factor of solar radiation, $\tau$ is the film transmissivity factor of solar radiation, and $r_{effective}$ is the solar radiation effective reflectivity factor of the balloon film.

$\tau_{atm}$ is specified as [9]

$$\tau_{atm} = 0.5 * \left[e^{-0.65 * Air_{Mass}} + e^{-0.95 * Air_{Mass}}\right], \tag{5}$$

where $Air_{Mass}$ is the absolute atmospheric optical quality. As the altitude increases, the radius of Earth's curvature changes, meaning that absolute atmospheric optical quality needs to be represented in two forms [21]:

- solar elevation angle $< 5°$:

$$Air_{Mass} = \left(\sin h + 0.457 * (90 - h)^{0.07}\right) * (6.484 + h)^{-1.697} * \left(\frac{P_{air}}{P_0}\right)^{(0.488 + 0.15 * h)} \tag{6}$$

- solar elevation angle $> 5°$:

$$Air_{Mass} = \left(\sin h + 0.457 * (90 - h)^{0.07}\right) * \left((6.484 + h)^{-1.697}\right)^{-1} * \frac{P_{air}}{P_0}, \tag{7}$$

where $h$ stands for the solar elevation angle, and $P_0$ is for the standard atmospheric pressure. The solar elevation angle varies with the region, season, and time of day, but it ranges from $0°$ to $90°$. For a specified location, the solar elevation angle equals $0°$ from sunset to sunrise. During the daytime, it reaches $90°$ at noon, which is the greatest intensity of solar radiation for the day. The solar elevation angle is formulated as

$$\sin h = \sin \varphi * \sin \zeta + \cos \varphi * \cos \zeta * \cos \omega, \tag{8}$$

where $\varphi$ refers to the local latitude, $\zeta$ refers to the solar declination angle, and $\omega$ refers to the solar hour angle. The variation of the solar declination angle is the main cause of the variation of solar radiation intensity on Earth's surface, which is formulated as [21]

$$\zeta = 23.45 * \sin\left(2 * \pi * \frac{Nday + 284}{365}\right), \tag{9}$$

where $Nday$ stands for the day's serial number, and it is computed from the first of January each year. The solar hour angle is defined as

$$\omega = (TH - 12) * 15, \tag{10}$$

where $TH$ represents the true solar time. There are two reasons causing the time difference between true solar time and mean solar time. First, the distance and the relative position between the Sun and Earth vary with time. Secondly, Earth's equator does not coincide with the plane of its orbit around the Sun. The true solar time is specified as [21]

$$TH = TH_s \pm \frac{La - La_s}{15} + \frac{\Delta TH}{60}, \tag{11}$$

where $TH_s$ stands for the regional standard time, which depends on which time zone the balloon flight system is in. $La$ stands for the local longitude, and $La_s$ refers to the regional standard time position's longitude, which is equivalent to the longitude of the centerline of the time zone. $\Delta TH$ stands for the time difference. The time difference is formulated as [21]

$$\Delta TH = -0.00167 + 7.36629 * \cos(W + 1.4981) - 9.92745 * \cos(2 * W - 1.26155) - 0.32123 * \cos(3 * W - 1.15710), \tag{12}$$

where $W$ represents Earth's orbit time angle and is indicated by the following formula [21]:

$$W = \frac{2 * \pi * Nday}{365} + 2 * e * \sin\left(\frac{2 * \pi * Nday}{365}\right) + 1.25 * e^2 * \sin\left(2 * \frac{2 * \pi * Nday}{365}\right), \tag{13}$$

where $e$ stands for Earth's orbit eccentricity. For use in the equations below, $r_{effective}$ is formulated as [10]

$$r_{effective} = r + r^2 + r^3 + r^4 + r^5 + \dots, \tag{14}$$

where $r$ stands for the solar radiation reflectivity factor of the balloon film.

### 2.2.2. Ground-Reflected Radiation

Ground-reflected radiation is specified as [9]

$$Q_{Albedo} = \alpha * \rho_{earth} * A_{surf} * I_{sun} * \sin h * \tau_{ViewFactor} * \left[1 + \tau * \left(1 + r_{effective}\right)\right], \tag{15}$$

where $\rho_{earth}$ indicates Earth's surface reflectance factor, $\tau_{ViewFactor}$ is the angular coefficient from the balloon's surface to Earth's surface, and $A_{surf}$ stands for the surface area of the zero-pressure balloon.

### 2.2.3. Sky-Scattered Radiation

Sky-scattered radiation is formulated as [22]

$$Q_{Scatter} = 0.5 * \alpha * A_{surf} * I_{sun} * \sin h * \tau_{ViewFactor} * \frac{Air_{Mass} * (1 - \tau_{atm})}{Air_{Mass}^{-1.4 * \ln \tau_{atm}}} * \left[1 + \tau * \left(1 + r_{effective}\right)\right]. \tag{16}$$

### 2.2.4. Earth's Infrared Radiation

Earth's infrared radiation is indicated by the following formula [22]:

$$Q_{IR,\ earth} = \alpha_{IR} * \varepsilon_{earth} * \sigma * T_{earth}^4 * A_{surf} * \tau_{ViewFactor} * \tau_{atm,IR} * \left[1 + \tau_{IR} * \left(1 + r_{effective,\ IR}\right)\right], \tag{17}$$

where $\alpha_{IR}$ represents the balloon film absorption factor of infrared radiation, $\varepsilon_{earth}$ refers to the ground's average infrared emissivity factor, $\sigma$ is the Stefan–Boltzmann constant, $\tau_{IR}$ is the balloon film transmissivity factor of infrared radiation, $r_{effective,IR}$ is the infrared radiation effective reflectivity factor of the balloon film, $T_{earth}$ represents Earth's surface temperature, and $\tau_{atm,IR}$ is the atmospheric transmissivity factor of infrared radiation.

### 2.2.5. Atmospheric Infrared Radiation

Atmospheric infrared radiation is formulated as [22]

$$Q_{IR,\ Sky} = \alpha_{IR} * \sigma * \varepsilon_{sky} * T_{sky}^4 * A_{surf} * (1 - \tau_{ViewFactor}) * \left[1 + \tau_{IR} * \left(1 + r_{effective,\ IR}\right)\right], \tag{18}$$

where $\varepsilon_{sky}$ stands for the atmospheric average infrared emissivity factor. $T_{sky}$ refers to the sky equivalent temperature, which is defined as [23]

$$T_{sky} = 0.052 * T_{air}^{1.5} \tag{19}$$

### 2.2.6. Surface Thermal Radiation

The infrared radiation of the balloon film includes the film's external surface and the film's internal surface's infrared radiation. The total surface thermal radiation is calculated by the following formula [24]:

$$Q_{IR,\ film} = Q_{IR,out} - Q_{IR,\ absorb} = \varepsilon * \sigma * 2 * A_{surf} * T_{film}^4 - \sigma * \varepsilon * \alpha_{IR} * A_{surf} * T_{film}^4 * \left(1 + r_{effectivce,\ IR}\right), \tag{20}$$

where the $\varepsilon$ stands for the average infrared emissivity factor of the film material, and $T_{film}$ represents the temperature of the film.

### 2.2.7. Convective Heat Transfer

The convective heat transfer of a high-altitude balloon includes the convective heat transfer between the balloon film and the external atmosphere and the convective heat transfer between the film and the internal helium.

For the convective heat transfer between the film and the external atmosphere, there are two types of convective modes: natural convection and forced convection. These are defined as [9]

$$Q_{ConvExt} = H_{external} * A_{surf} * \left(T_{air} - T_{film}\right), \tag{21}$$

where $H_{external}$ represents the external convection heat transfer coefficient. For the convection heat transfer between the film and the external atmosphere, it is usually a coupled heat transfer process, so $H_{external}$ is specified as [10]

$$H_{external} = \left(H_{force}^3 + H_{free}^3\right)^{1/3}, \tag{22}$$

where $H_{force}$ represents the forced convection heat transfer coefficient, and $H_{free}$ is the natural convection heat transfer coefficient.

For the convection heat transfer between the film and the internal helium, its convective mode is natural convection, which is specified as [9]

$$Q_{ConvInt} = H_{internal} * A_{surf} * \left(T_{film} - T_{helium}\right), \tag{23}$$

where $T_{helium}$ is the temperature of the helium.

#### 2.2.8. Heat Transfer on Balloon Film

In this study, the temperature of the film is assumed to be uniform, and it is formulated as [9]

$$\frac{dT_{film}}{dt} = \frac{Q_{Direct} + Q_{Albedo} + Q_{Scatter} + Q_{IR,\,earth} + Q_{IR,\,sky} + Q_{ConvExt} - Q_{ConvInt} - Q_{IR,\,film}}{C_f * m_{film}}, \tag{24}$$

where $C_f$ is the specific heat capacity of the film material, $Q_{Direct}$ stands for direct solar radiation, $Q_{Albedo}$ is ground-reflected radiation, $Q_{Scatter}$ is sky-scattered radiation, $Q_{IR,earth}$ is Earth's infrared radiation, $Q_{IR,sky}$ stands for atmospheric infrared radiation, $Q_{ConvExt}$ and $Q_{ConvInt}$ represent convective heat transfer, and $Q_{IR,film}$ refers to surface thermal radiation.

#### 2.2.9. Heat Transfer on Internal Helium

This study also assumes that the temperature and pressure of the internal helium are uniform. According to the first law of thermodynamics, the temperature change rate of the internal helium is specified as [9]

$$\frac{dT_{helium}}{dt} = \frac{Q_{ConvInt}}{C_v * m_{helium}} + (\gamma - 1) * T_{helium} * \left( \frac{dm_{helium}}{dt} * \frac{1}{m_{helium}} - \frac{dVolume_{helium}}{dt} * \frac{1}{Volume_{helium}} \right), \tag{25}$$

where $C_v$ represents the specific heat capacity at a constant volume of helium. Here, $\gamma$ means the specific heat ratio, which equals $C_p/C_v$. $C_p$ is the specific heat capacity at a constant pressure of helium.

### 2.3. Earth Model

As the balloon flight system moves, the local Earth radius and gravity change with its location. In this study, the WGS-84 Earth model is used to offer real-time data on Earth's radius and the acceleration of gravity at the current balloon location. The corresponding equations are defined as [25]

$$R_{meridian} = \frac{r_e * (1 - \epsilon^2)}{\left(1 - \epsilon^2 * (\sin \varphi)^2\right)^{3/2}} \tag{26}$$

$$R_{normal} = \frac{r_e}{\left(1 - \epsilon^2 * (\sin \varphi)^2\right)^{1/2}} \tag{27}$$

$$R_{equiv} = \sqrt{R_{meridian} * R_{normal}} \tag{28}$$

$$g = g_{WGS_0} \frac{1 + g_{WGS_1} * (\sin \varphi)^2}{\left(1 - \epsilon^2 * (\sin \varphi)^2\right)^{1/2}}, \tag{29}$$

where $R_{meridian}$ stands for meridian radius, $R_{normal}$ means normal radius, $R_{equiv}$ refers to equivalent radius, $g$ is the acceleration of gravity, $r_e$ stands for equatorial radius, $\epsilon$ represents first eccentricity, $g_{WGS_0}$ is gravitational acceleration at the equator, and $g_{WGS_1}$ is the gravity formula constant.

## 3. Trajectory Prediction Model

To simulate the motion of the balloon, the following hypotheses are recognized:

(1)　The horizontal components of the balloon's speed are equal to the horizontal components of the wind's speed.

(2)　The balloon is assumed to be a point mass when considering the external forces acting on the balloon.

(3)　The high-altitude wind speed magnitude and direction are assumed to be constant for at least 24 h.

### 3.1. Wind Model

As the altitude increases, the wind speed's magnitude and direction change with it. Wind direction is usually divided into latitudinal wind and longitudinal wind. For latitudinal wind, the wind blowing from the west is positive, while for longitudinal wind, the wind blowing from the south is positive.

In this study, the wind data are measured on August 8, 2019 (UTC+16) in China, which include the wind speed magnitude and direction from the ground to 32 km in altitude. After data processing, these discrete wind data are fitted to a series of continuous curves by the sum of sines models, which is specified as [26]

$$y = \sum_{i=1}^{n} a_i * \sin(b_i * x + c_i), \tag{30}$$

where $a$ represents amplitude, $b$ refers to frequency, and $c$ is the phase constant for each sine wave term. $n$ is the number of terms in the series, and $1 \leq n \leq 8$. Figure 4 shows the discrete wind data and the corresponding fitted curve.

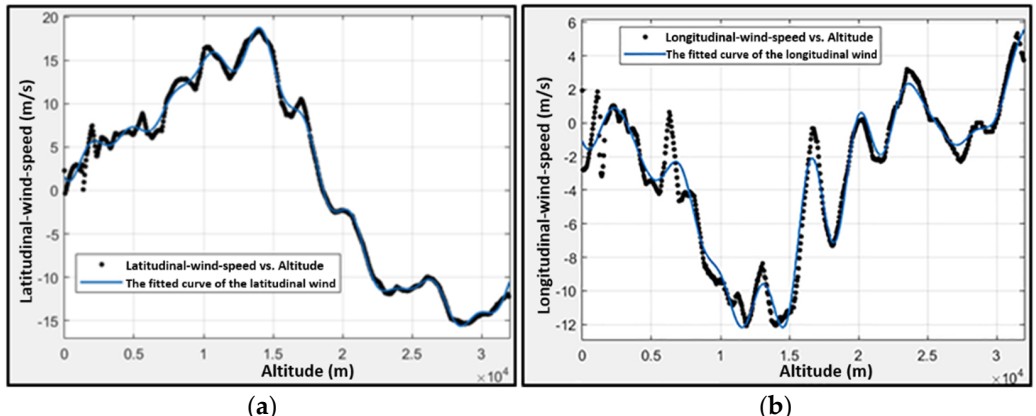

(a)                                                    (b)

**Figure 4.** Discrete wind data and the corresponding fitted curve. (**a**) shows the discrete latitudinal wind data and the fitted curve of the latitudinal wind. (**b**) shows the discrete longitudinal wind data and the fitted curve of the longitudinal wind.

Based on these fitting results, the corresponding fitted curves are defined as

(1)   the fitted curve of the latitudinal wind speed (m/s) vs. altitude (m):

$$f(z)_{latitude} = \sum_{i=1}^{8} a_i * \sin(b_i * z + c_i), \tag{31}$$

where $z$ indicates altitude, $f(z)_{latitude}$ is the latitudinal wind speed at the corresponding altitude, $a_i = [23.39, 30.97, 7.256, 35.02, 1.212, 1.159, 34.26, 0.9401]$, $b_i = [7.183e-05, 3.544e-05, 0.00031, 0.00099, 0.001989, 0.000675, 0.000992, 0.00165]$, and $c_i = [0.4929, 3.151, 3.723, 5.658, -1.421, -2.213, 2.584, -2.243]$; and

(2)   the fitted curve of the longitudinal wind speed (m/s) vs. altitude (m):

$$f(z)_{longitude} = \sum_{i=1}^{8} a_i * \sin(b_i * z + c_i), \tag{32}$$

where $z$ indicates altitude, $f(z)_{longitude}$ is the longitudinal wind speed at the corresponding altitude, $a_i = [80.21, 3.932, 80.58, 1.416, 1.007, 0.2995, 0.8878, 1.157]$, $b_i = [8.839e-05, 0.00036, 9.238e-05, 0.0018, 0.00078, 0.00123, 0.0029, 0.001476]$, and $c_i = [-0.7307, 0.3148, 2.454, -2.866, 1.572, -0.4936, -1.095, 2.805]$.

### 3.2. Geometry Model

A zero-pressure balloon with a shape factor of 0.2 is employed as the research object. The detailed geometric equations are as follows [9]:

(1)   diameter from the top view:

$$Diameter = 1.383 * Volume^{1/3} \tag{33}$$

(2)   projected area from the top view:

$$A_{top} = \frac{\pi}{4} * Diameter^2 \tag{34}$$

(3)   During the balloon flight process, the illuminated projected area varies with the solar elevation angle, which is defined as

$$A_{projected} = A_{top} * [0.9125 + 0.0875 * \cos(\pi - 2 * h)] \tag{35}$$

(4)   surface area of the zero-pressure balloon:

$$A_{surf} \approx 4.94 * Volume^{2/3} \tag{36}$$

(5)   When the balloon is not fully inflated, the surface of the balloon is presented in crenellated form. The maximum crenellated surface area is defined as follows:

$$A_{surf1} = 4.94 * Volume_{design}^{2/3} * \left(1 - \cos\left(\pi * L_{goreB}/L_{goreDesign}\right)\right), \tag{37}$$

where $L_{goreB}$ indicates the actual length of gore exposed in the balloon, $L_{goreDesign}$ indicates the designed gore length of the balloon, and $Volume_{design}$ indicates the maximum volume of the balloon.

(6)   To better approximate a realistic situation for convection and film mass calculations, $A_{effective}$ is used to describe the effective exposed surface area:

$$A_{effective} = 0.65 * A_{surf} + 0.35 * A_{surf1} \tag{38}$$

(7)   A zero-pressure balloon has a distinctive feature: the internal helium pressure is almost equal to the surrounding atmospheric pressure. The average pressure difference between the internal helium pressure and the surrounding atmospheric pressure is defined as

$$\Delta P = 0.517 * g * (\rho_{air} - \rho_{helium}) * Volume^{1/3}, \tag{39}$$

where $\rho_{helium}$ is the density of helium.

Furthermore, based on the ideal gas law, the balloon's volume can be formulated as

$$Volume = \frac{m_{helium} * R_{helium} * T_{helium}}{P_{helium}} \tag{40}$$

$$P_{helium} = P_{air} + \Delta P, \tag{41}$$

where $Volume = Volume_{balloon} = Volume_{helium}$, $R_{helium}$ stands for the helium gas constant, $T_{helium}$ represents helium temperature, $P_{helium}$ refers to helium pressure, and $P_{air}$ is the surrounding atmosphere pressure.

*3.3. Dynamic Model*

The motion of a zero-pressure balloon is divided into vertical motion and horizontal motion. For the vertical motion, the gross external force consists of buoyant force, gross weight, and aerodynamic drag force. The vertical dynamic equation of motion is defined as [27]

$$m_{virtual} * \frac{dv_z}{dt} = (\rho_{air} - \rho_{helium}) * Volum * g - m_{gross} * g - \frac{1}{2} * C_d * \rho_{air} * A_{top} * v_z^2, \tag{42}$$

where $v_z$ is the vertical speed of the balloon flight system; $m_{gross}$ is the gross mass of the payload, film, and ballast; $m_{virtual}$ is the gross mass of the payload mass, film mass, ballast mass, helium mass, and air virtual mass. Air virtual mass is specified as

$$m_{air,\ virtual} = C_{vir} * \rho_{air} * Volume_{helium},\tag{43}$$

where $C_{vir}$ stands for the virtual mass coefficient, and $C_{vir} \approx 0.5$. $C_d$ is the drag coefficient, which is defined as [28]

$$\begin{cases} C_d = 1.5 * \left( \frac{24}{Re} + \frac{6}{1+\sqrt{Re}} + 0.4 \right), \quad Re \leq 2.7 * 10^5 \\ \log C_d = 1.5 * (25.821 - 4.825 * \log Re), \quad 2.7 * 10^5 < Re \leq 3.7 * 10^5 \\ \log C_d = 1.5 * \left( -0.699 - 0.347 * e^{-38.533 * (\log \frac{Re}{3.7 * 10^5})^{5.306}} \right), \quad 3.7 * 10^5 < Re \leq 10^6 \\ C_d = 0.3, \quad Re > 10^6 \end{cases}\tag{44}$$

For the horizontal motion, the balloon is regarded as moving with the wind, assuming that the horizontal velocity is equal to the wind speed's magnitude. The horizontal dynamic equation is defined as

$$v_{east} = \frac{dx_{latitude}}{dt} = v_{wind} * \sin \theta\tag{45}$$

$$v_{north} = \frac{dx_{longitude}}{dt} = v_{wind} * \cos \theta,\tag{46}$$

where $v_{east}$ and $v_{north}$ are the longitudinal and latitudinal balloon speed, respectively, and $x_{latitude}$ and $x_{longitude}$ are the longitudinal and latitudinal distance, respectively. Here, $\theta$ indicates the angle between the direction of the horizontal wind and due north. In Earth's coordinate system, the balloon's horizontal displacements are defined as [29]

$$\dot{Lat} = \frac{v_{north}}{R_{meridian} + ALT}\tag{47}$$

$$\dot{Lon} = \frac{v_{east}}{(R_{normal} + ALT) * \cos Lat},\tag{48}$$

where $ALT$ indicates altitude. $Lat$ and $Lon$ are the latitude and longitude, respectively.

*3.4. Exhaust Model*

3.4.1. Automatic Exhaust Model

The zero-pressure balloon's ascending process includes two phases: one with the balloon partially expanded and the other with the balloon fully expanded.

The balloon fully expanded means that the balloon has expanded to its designed volume, then the overfilled helium is automatically vented from the exhaust pipes installed at the bottom of the balloon, preventing the balloon from exploding. This helium exhaust process is called the automatic exhaust process. The corresponding helium mass loss is defined as

$$\frac{dm_{helium}}{dt} = -\rho_{helium} * \frac{Volume_{altitude} - Volume_{design}}{dt},\tag{49}$$

where $\rho_{helium}$ stands for the density of the helium, and $Volume_{altitude}$ is the balloon volume with the corresponding altitude. The equation works on the premise that the balloon has expanded to its designed volume.

3.4.2. Active Exhaust Model

Considering realistic flight mission requirements and some emergencies, the balloon flight system needs to be forced to descend. To accomplish this, a helium exhaust valve installed at the apex of the balloon can be opened to exhaust helium. This helium exhaust

process is called the active exhaust process. The corresponding helium mass loss is specified as [30]

$$\frac{dm_{helium}}{dt} = -\rho_{helium} * c * A * \sqrt{\frac{2 * \Delta P}{\rho_{helium}}}, \tag{50}$$

where $c$ indicates a helium flow coefficient that depends on the number and shape of the helium exhaust valves, and $A$ is the total opening area of the exhaust valves.

### 3.5. Ballast Model

After local noon, the intensity of solar radiation gradually weakens, causing the internal helium temperature to drop, and the volume of the balloon starts to shrink. Eventually, the zero-pressure balloon flight system begins to descend. To keep the designed altitude stable or to control the vertical landing velocity, a corresponding mass of ballast needs to be dropped.

In this study, granular iron sand is used as ballast, and the ballast discharge rate is determined experimentally by dropping ballast before the balloon flight mission. The ballast model is defined as

$$m_{ballast_{remained}} = m_{ballast_{actual}} - \int_0^t k_{ballast} dt, \tag{51}$$

where $m_{ballast_{actual}}$ stands for the original mass of the ballast, $m_{ballast_{remained}}$ refers to the mass of the ballast after dropping, and $k_{ballast}$ is the ballast flow rate.

## 4. Results and Discussion

The simulation program used Matlab/Simulink, and the corresponding simulation program structure diagram is shown in Figure 5. Figure 5 shows the above models, including the atmospheric, thermal, Earth, wind, geometry, exhaust, dynamic, and ballast models. The simulation process was realized through the data flow among the models, and the nonlinear differential equations were solved by the Runge–Kutta methods.

### 4.1. Model Validation

The best way to evaluate the simulation program was to compare the simulation results with real flight data. However, the flight mission data, balloon specification, atmospheric information, and film surface characteristics were almost inaccessible. In this paper, NASA's scientific balloon analysis model (SINBAD) and Palumbo et al.'s analysis model for high-altitude balloons (ACHAB) were used to validate the present model [31,32]. Both models have been proven to offer highly accurate trajectory predictions and were compared with real flight data [14,32]. The corresponding input data and output data can be found in [32]. Based on the previous works, the drag coefficient was taken into consideration for model validation. The varying drag coefficient was used for ACHAB and the present model, which was defined by the corresponding functions. A constant drag coefficient of 0.45 was used for ACHAB, SINBAD, and the present model. The simulation results are shown in Figure 6.

On the left side of Figure 6, the vertical velocity profile of the present model seems to be similar to that of ACHAB and SINBAD with the constant drag coefficient. The reason for the differences at the initial time and the time of reaching the designed altitude was that the ground surface temperature model and atmospheric model were different, which affected the ascent flight performance. For the varying coefficient, the time of reaching the designed altitude for both models was the same. All of the velocity profiles took on a W shape prior to float. On the right side of Figure 6, the float altitude of the present model is slightly lower than the previous programs. The reason for this was that atmosphere density and gravity acceleration differences mainly affected the net buoyancy at the designed altitude. In the present model, the variation of gravity due to the position was considered, while for the other models, it was not considered.

In summary, the present model appeared to be similar to the previous simulation programs and could offer enough accuracy for trajectory prediction. However, we still needed to optimize the simulation program with an interpolation model that considers the relationship between height and drag coefficient, since the balloon's geometry changes with the altitude during the ascent phase. Furthermore, to improve its accuracy, the simulation model needs to be further optimized by the flight test data in the following research.

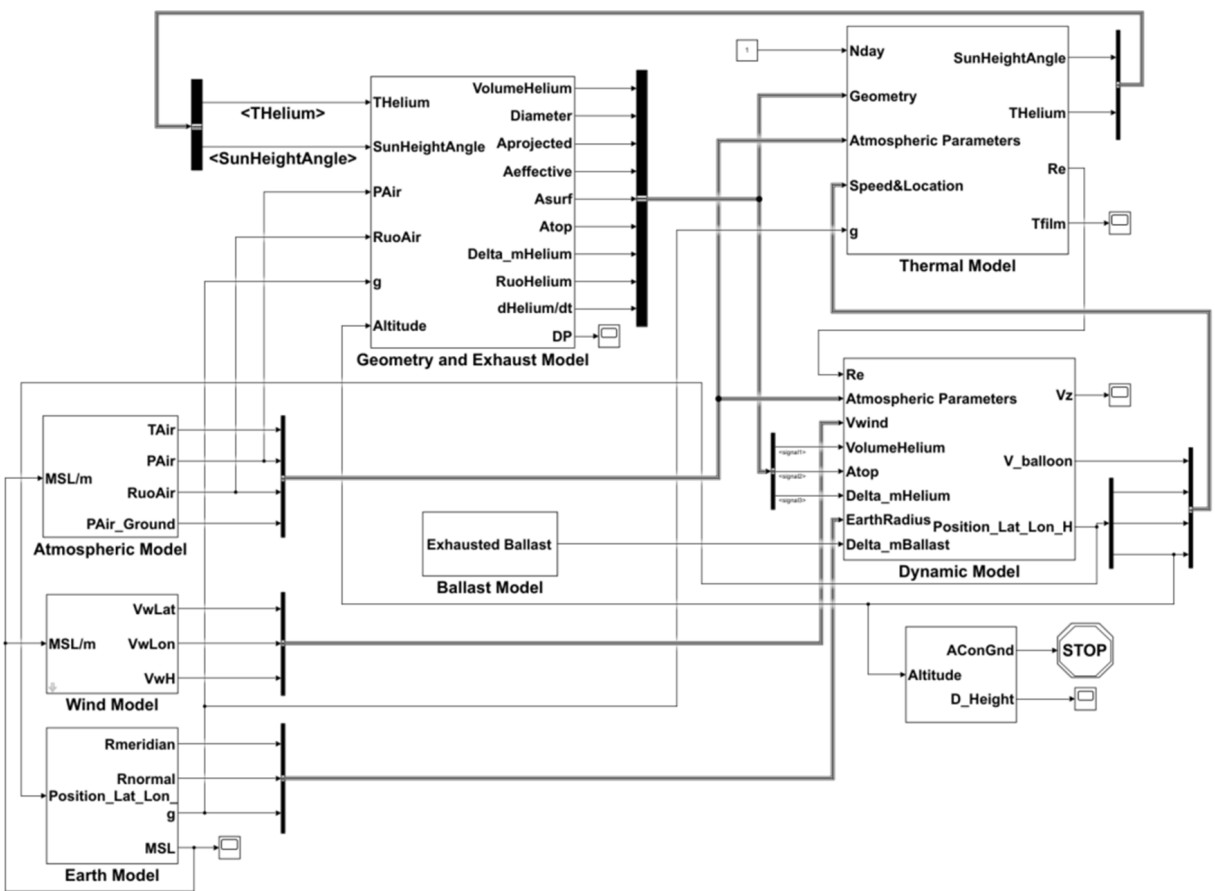

**Figure 5.** Matlab/Simulink simulation program structure diagram.

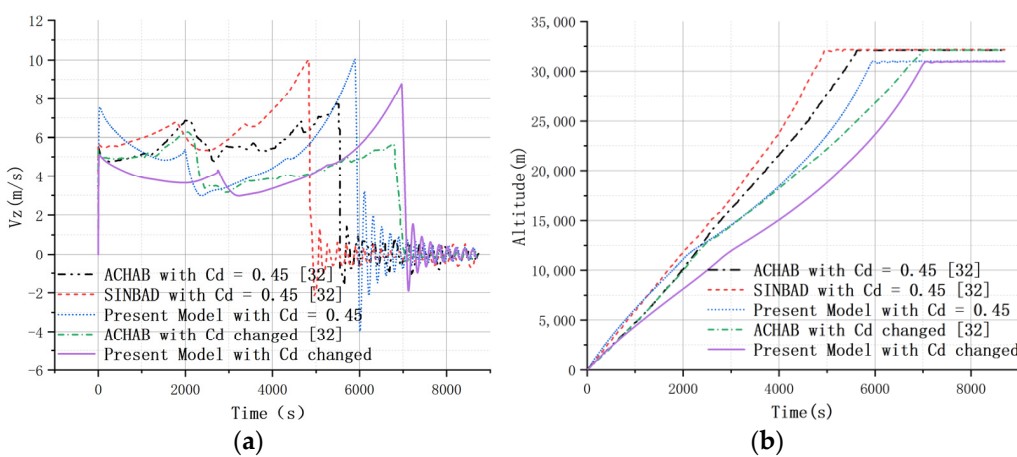

**Figure 6.** Simulation results. (**a**) shows the comparison of vertical speed vs. time. (**b**) shows the comparison of altitude vs. time.

### 4.2. Flight Simulation

The zero-pressure balloon's specifications and the corresponding flight simulation information are presented in Table 1. The flight performance was simulated during the entire trajectory in detail.

**Table 1.** Balloon's specifications and flight simulation information.

| Parameter | Value |
|---|---|
| Maxi volume | 44,930.5 m$^3$ |
| Lifting gas (helium) mass | 626.5 kg |
| Balloon mass | 487 kg |
| Suspending mass (payload) | 2930.87 kg |
| Initial film temperature | 286.95 K |
| Initial helium temperature | 286.95 K |
| Number of the helium exhaust valves | 2 |
| Film absorption factor of solar radiation | 0.33 |
| Film transmissivity factor of solar radiation | 0.65 |
| Film absorption factor of infrared radiation | 0.75 |
| Film transmissivity factor of infrared radiation | 0.20 |
| Film average infrared emissivity factor | 0.75 |
| Earth surface reflectance factor | 0.3 |
| Ground average infrared emissivity factor | 0.95 |
| Launch date and time | 8 August 2019 (UTC +16) |
| Launch base | Lat: 39°, Lon: 105° |
| Target altitude | 20 km |
| Ground air pressure | 101,325 Pa |
| Ground air temperature | 288.15 K |

UTC is the abbreviation for Universal Time Coordinated.

The variations of film temperature, internal helium temperature, atmospheric temperature, altitude, exhausted helium mass, vertical velocity, and trajectory throughout the flight mission are shown in Figures 7–10.

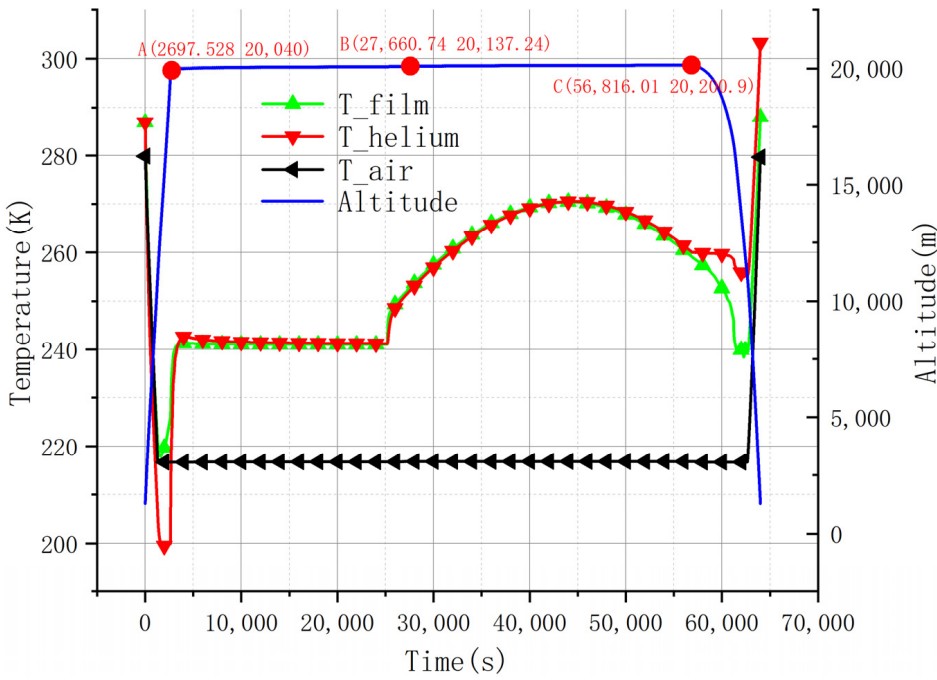

**Figure 7.** Temperature change of the film, helium, and air vs. time, and altitude change vs. time.

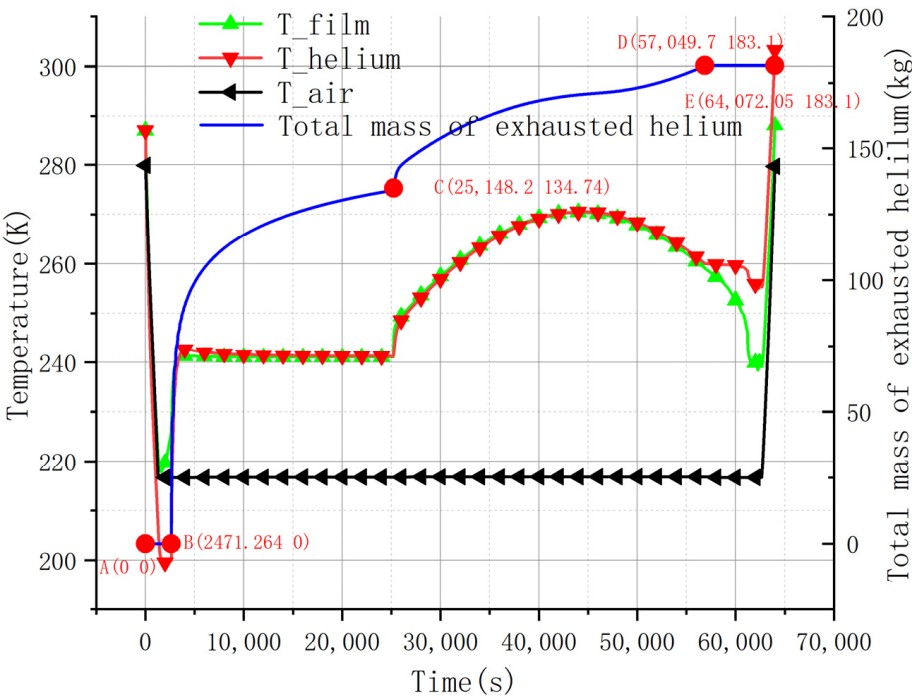

**Figure 8.** Temperature change of the film, helium, and air vs. time, and the change of the total mass of exhausted helium vs. time.

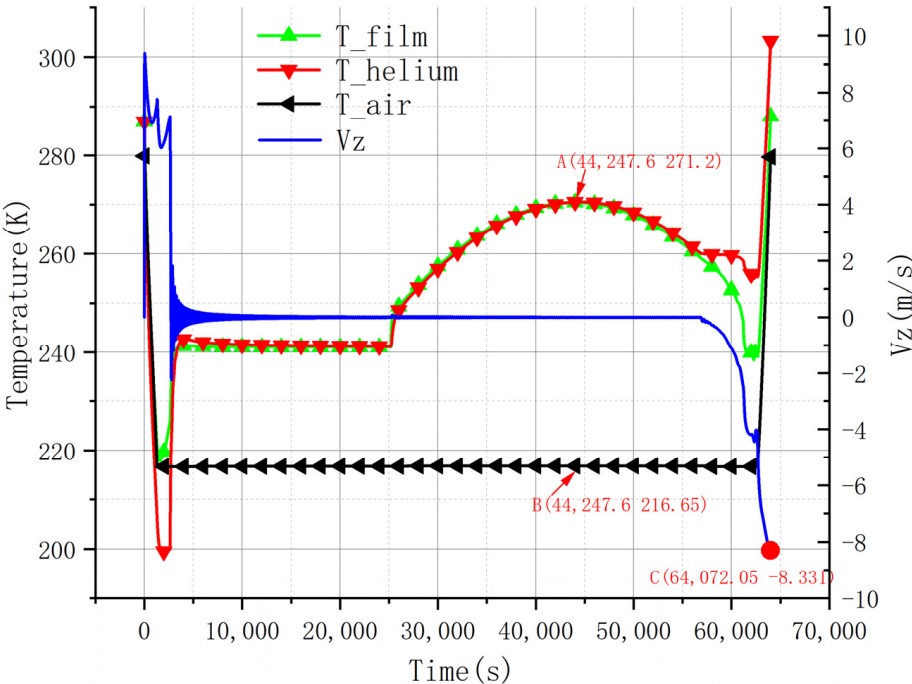

**Figure 9.** Temperature change of the film, helium, and air vs. time, and the change of the vertical speed of the balloon flight system vs. time.

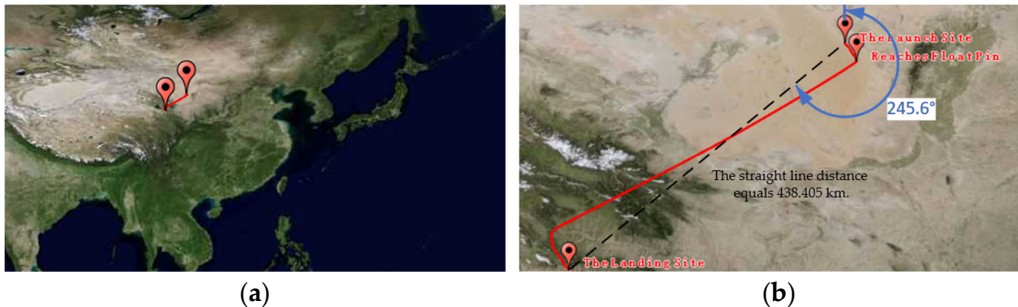

**Figure 10.** Trajectory of the balloon flight system. (**a**) shows the relative position of the trajectory on the world map. (**b**) shows the trajectory in detail.

1. Change Pattern of Temperature

During the ascending phase, the temperatures of the film and helium decreased simultaneously with the atmospheric temperature. In addition, the temperatures of both the film and helium were lower than the atmospheric temperature, causing the phenomenon of supercooling. The maximum temperature difference reached $-17.2°C$ between the helium and the surrounding atmosphere. The reason for this was that the volume of the balloon expanded rapidly, and the internal helium did positive work on the external atmosphere.

After the balloon flight system reached the designed float altitude, the temperatures of the film and helium gradually increased and came to equilibrium with the ambient thermal environment. According to Figure 7, the trend of temperature increase in the balloon film and internal helium became stable, and the increase patterns tended to be similar. After sunrise, the internal helium and film temperatures rose rapidly again due to solar radiation, with the temperature difference peaking at noon. The corresponding temperature difference between the helium and the surrounding atmosphere was $+54.55°C$.

In the afternoon, the temperature of the internal helium began to drop as the solar radiation decreased, resulting in a decrease in balloon volume and net buoyancy. The balloon began to move downward. During the descent, the external atmosphere did work on the helium expansion together with solar radiation, effectively compensating for the surface of the balloon so that helium temperature changed very little in the stratospheric range. After entering the tropospheric range, the temperature of the helium rose rapidly, causing the phenomenon of superheating. By the time the balloon system reached the ground, the helium temperature could reach $43°C$.

2. Change Pattern of Altitude

The ascent and descent processes of the balloon flight system were manipulated by the vertical velocity. As shown in Figures 9 and 11, the ascent time of the system was almost 75 min faster than the descent process. In addition, during the floating phase, the balloon reached its designed volume. The increase in helium temperature caused the internal helium to expand, and a corresponding amount of helium was continuously discharged from the bottom exhaust pipe, causing the balloon flight system to move upward due to the buoyancy force being greater than gravity. The maximum altitude eventually reached 20,200.9 m. After reaching that altitude, the balloon flight system started to move downward.

3. Change Pattern of Vertical Speed

Throughout the ascent process, the vertical velocity changed in a W pattern. Initially, the balloon flight system rose rapidly, and its vertical velocity increased sharply to 9.285 m/s. Then, the vertical velocity decreased due to the weakening of the upward net buoyancy caused by the supercooling properties of the internal helium. After reaching a certain altitude, the atmospheric density was reduced to a minimum magnitude, resulting in the drag going down to a relatively small value and the vertical speed increasing.

When the balloon flight system climbed above 11 km, the vertical velocity decreased significantly and then increased again. Furthermore, when the system was about to reach the designed altitude, the vertical velocity dropped sharply and oscillated around 0 m/s until it finally dropped gradually to 0 m/s. In the afternoon, influenced by the surrounding thermal environment, the balloon flight system started to descend, and its downward vertical velocity continuously increased, reaching 8.331 m/s by the time it reached the ground.

4.  Change Pattern of the Total Exhausted Mass of Helium

A high-altitude zero-pressure balloon has a unique and distinctive feature, which is exhaust pipes installed at the bottom of the balloon for exhausting helium. This is the reason why a zero-pressure balloon flight system cannot float for a long time. Therefore, it was imperative to figure out the helium-exhausting process in detail to plan the flight mission.

From the results described in Figures 7 and 9, the ascent process could be divided into two phases: the non-forming and forming geometry phases. The non-forming geometry phase occurred mainly at the initial time of the ascent process. As the external atmospheric pressure decreased, the volume of the balloon kept expanding while the helium mass remained constant. When the balloon flight system was about to reach the designed altitude, the forming geometry phase appeared. In this phase, the volume of the balloon reached its maximum and remained unchanged. Due to the momentum and vertical velocity, the balloon flight system continued to rise, causing excess helium to be discharged from the corresponding tubes. The helium mass began to decrease.

During the floating phase, the loss of helium mass continued, which was proportional to the solar radiation. After the balloon flight system began to descend, the balloon started to shrink, but the helium mass stayed constant until it reached the ground. During the entire flight process, the total exhausted mass of helium was 183.1 kg.

5.  Trajectory Prediction

As shown in Figure 10, the straight-line distance between the launch site and the landing site was 438.405 km, and the azimuth between the launch site and the landing site was 245.6°. Moreover, the flight mission lasted for nearly 18 h. The high-altitude zero-pressure balloon flew in a large volume of airspace without a corresponding control strategy, and the trajectory took on a Z shape for this flight.

Based on these results, the balloon's performance during the entire flight mission could be discussed in detail, helping the balloon system's design and operation. Furthermore, it could be concluded that the vertical speed during landing has a serious impact on the safe recovery of a balloon flight system, but the flight time and flight distance are not conducive to its rapid recovery. Thus, it was imperative to better understand the impact of the uncertainty of the launch parameters on trajectory prediction and to develop the corresponding control strategies to assist with rapid recovery and achieve the efficient and safe application of a high-altitude zero-pressure balloon flight system in limited airspace.

4.2.1. Launch Time

Different launch times, including UTC + 16 on 8 August 2019 (Chinese standard time 00:00 on 9 August), UTC + 22 on 8 August 2019 (Chinese standard time 06:00 on 9 August), and UTC + 04 on 9 August 2019 (Chinese standard time 12:00 on 9 August), were used to simulate the flight mission. As shown in Figures 11 and 12, the launch time can play an important role in trajectory prediction. This is because direct solar radiation, ground-reflected radiation, sky-scattered radiation, and other thermal radiation vary periodically, causing a great difference in radiation for flights launched at different times of the day.

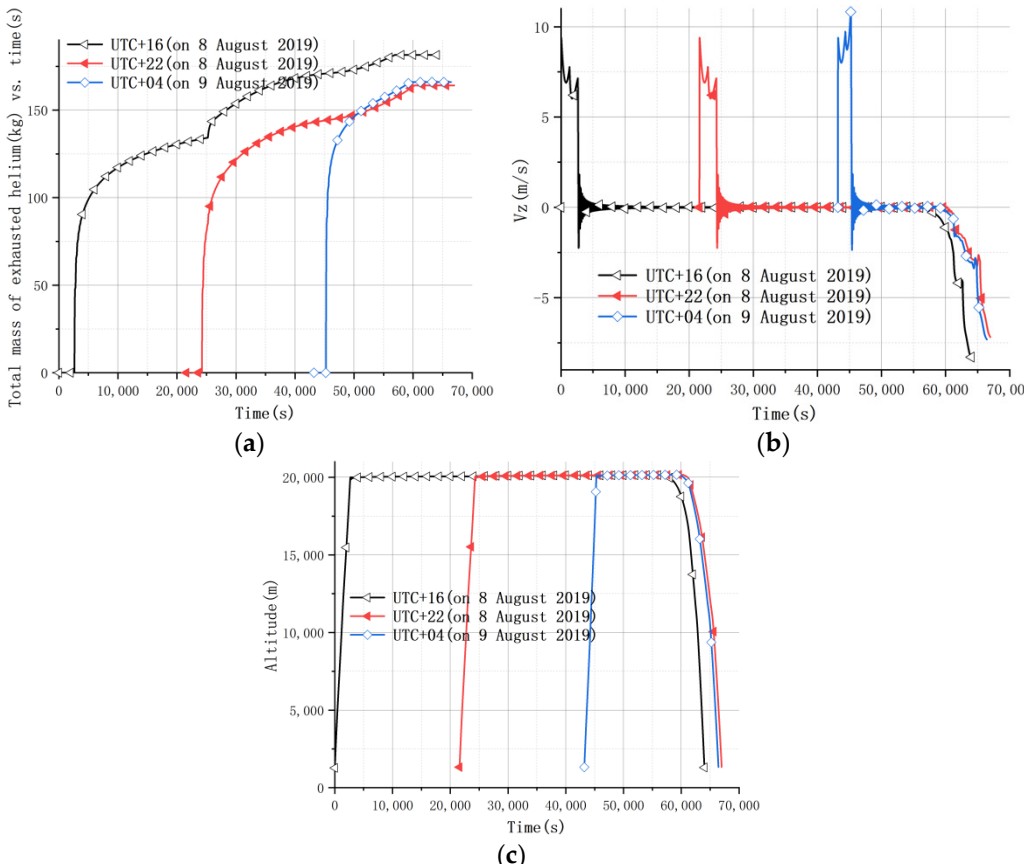

**Figure 11.** Diagrams corresponding to different launch times. (**a**) shows the total mass of exhausted helium vs. time, (**b**) shows the change in vertical velocity, and (**c**) shows the change in altitude.

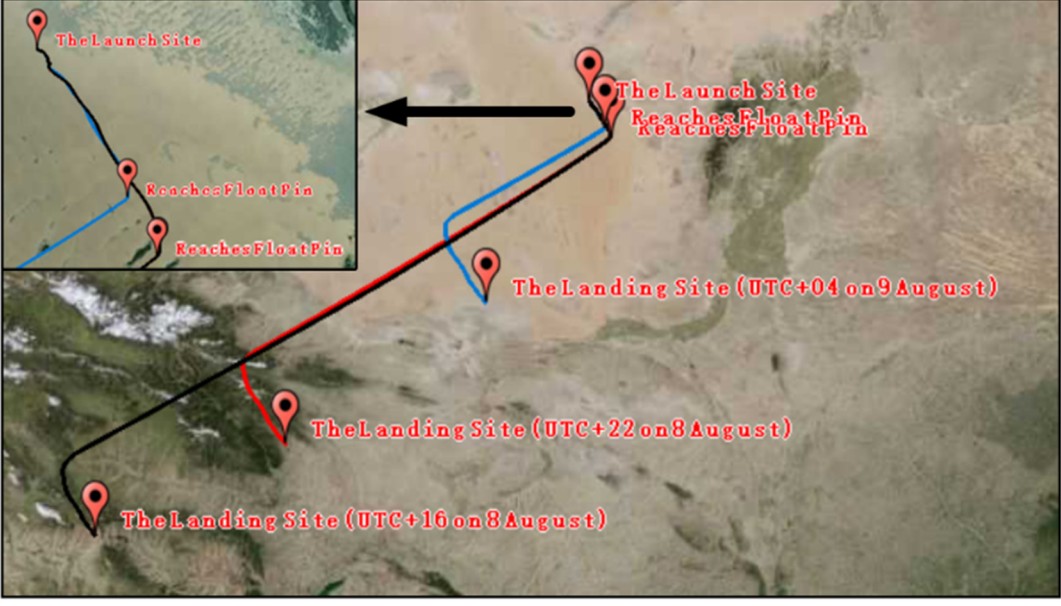

**Figure 12.** Trajectory predictions at different launch times of the day.

By comparing the flight performance of the zero-pressure balloon system at different times of the day, the following conclusions could be made:

1. The earlier the launch time, the more the total mass of helium is exhausted finally.

2. The earlier the launch time, the smaller the vertical velocity during the accent phase, resulting in relatively low requirements for balloon structure design. This is because the faster the vertical speed, the more severe the supercooling phenomenon.
3. The launch time has almost no effect on the floating altitude and landing vertical velocity.
4. The earlier the launch time, the longer the flight time and flight distance. However, the maximum flight time is less than 24 h.

Using these results, a suitable launch time can be chosen to start a flight mission and meet the mission requirements.

### 4.2.2. Initial Helium Mass

Different initial helium masses, such as 626 kg, 676 kg, and 726 kg, were used to simulate the flight mission on 8 August 2019 (UTC + 16). As shown in Figures 13 and 14, the corresponding results were as follows:

1. The more the initial mass of helium, the more the mass of helium was exhausted. However, the mass of helium inside the balloon was the same when landing.
2. The more the initial mass of helium, the faster the balloon system ascended.
3. The initial mass of helium had no effect on the floating altitude and trajectory prediction.

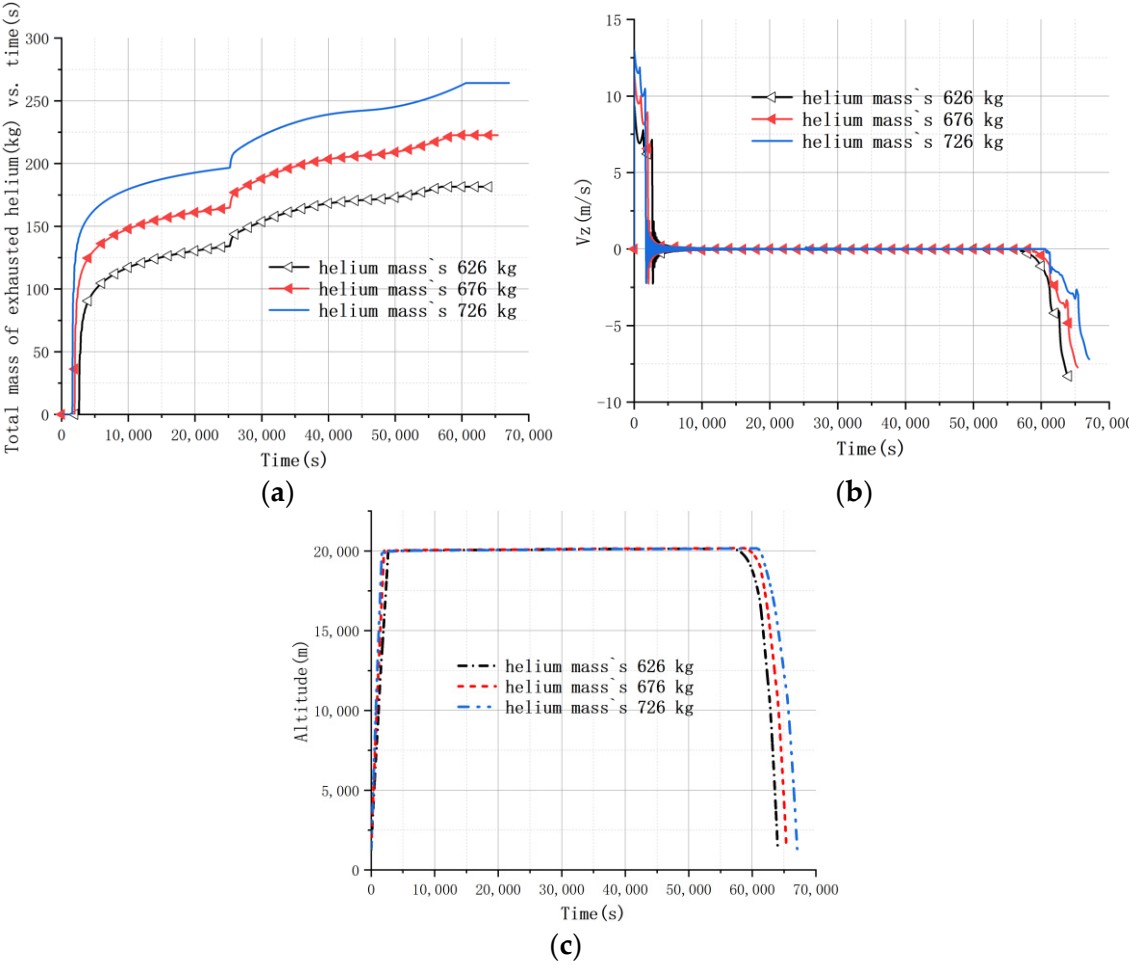

**(a)**

**(b)**

**(c)**

**Figure 13.** Diagrams corresponding to different initial masses of helium. (**a**) shows the total mass of exhausted helium vs. time, (**b**) shows the change in vertical velocity and (**c**) shows the change in altitude.

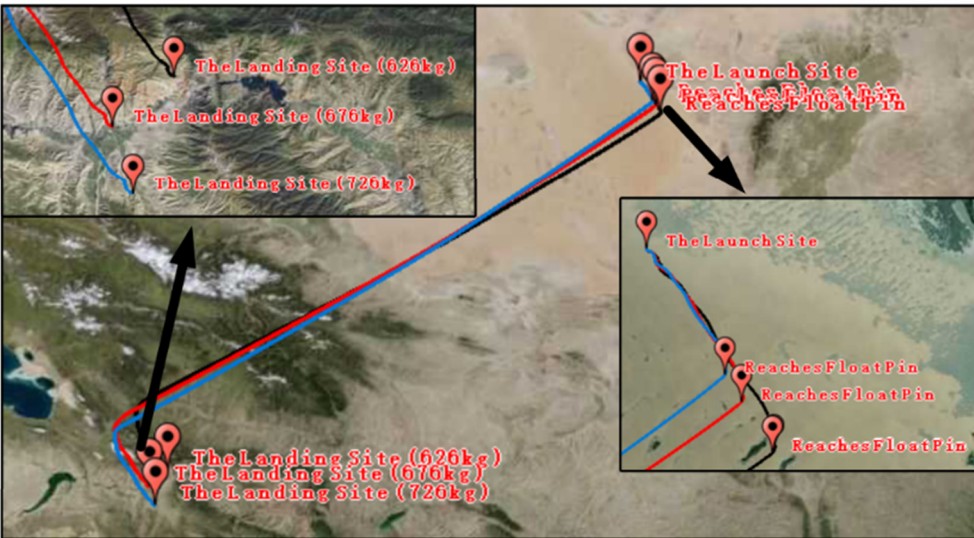

**Figure 14.** Trajectory predictions for different initial helium weights.

These results showed that the initial mass of helium could affect the flight performance during the ascent phase, increasing the ascent rate and decreasing the time needed to reach float, but it had no effect on the floating and descending phases. Therefore, the selection of the initial helium mass can be determined by the desired ascent velocity.

*4.3. Trajectory Prediction with Control Strategy*

These results showed that a zero-pressure balloon flight system flying in a huge amount of airspace makes it difficult to recover the balloon system, even considering the launch time. Furthermore, the landing speed of the balloon system is detrimental to safe recovery. Based on these considerations, the corresponding control strategy was employed to plan the trajectory.

In this study, the active exhaust model and ballast model were used together to control the trajectory. The control strategy also took the floating time and wind speed profile into consideration, which is helpful for real flight. The control strategies were as follows:

Most of the balloon flight missions only needed to float at the designed altitude for a few hours, sometimes less. It was clear that the zero-pressure balloon was in the automatic exhaust phase while in its float phase. Then, the balloon system began to use the active exhaust procedure to accelerate the transition to descent after finishing the floating mission. Once the balloon system started to descend, the vertical speed increased. Figure 15 shows the wind speed profile measured on 8 August 2019. This diagram illustrates that the horizontal wind speed was excessively large from an altitude of 10 km to an altitude of 15 km, causing excessive horizontal motion of the balloon system while passing through that altitude range. There was no doubt that traversing these height layers as quickly as practicable was reasonable. After the active exhaust procedure was finished, the landing speed needed to be optimized by the ballast model to ensure a safe landing.

In summary, the control strategies comprised two phases. First, enforcing the active exhaust procedure allowed the balloon system to descend until the designed ballast drop altitude, and then the active exhaust process was finished. Second, it was imperative to start dropping ballast immediately at the end of the active exhaust process, which helped the balloon system reach the ground at a vertical speed close to zero.

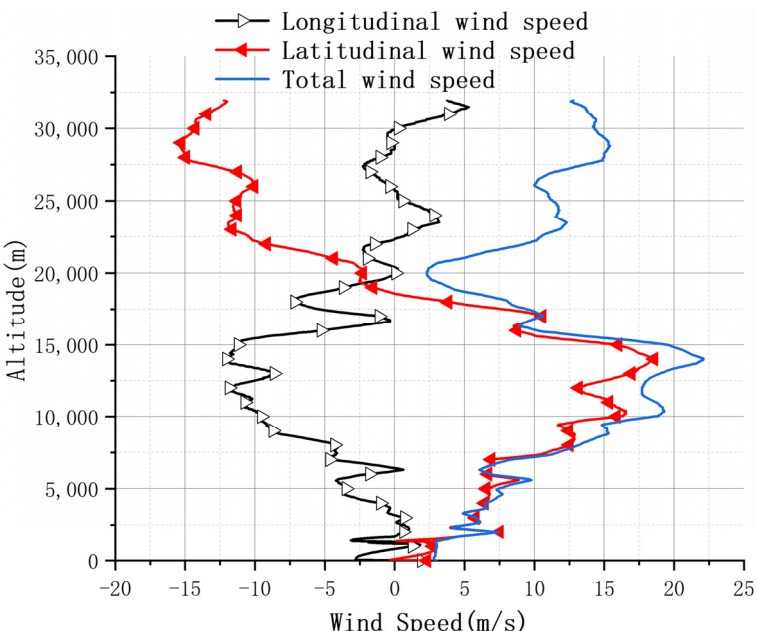

**Figure 15.** Wind speed profile vs. altitude on launch day.

Taking the floating mission time, horizontal motion, and landing speed into consideration, the results from the balloon flight system's trajectory simulation were as follows:

1. With the control strategy applied, the flight time and lateral drift distance were greatly reduced. For Mission A, the straight-line distance was 63.17 km. For Mission B, the straight-line distance was 62.28 km, and for Mission C, the straight-line distance was 70.46 km. For Mission D, the straight-line distance was 69.61 km. For all the flight missions, the flight time was less than 3 h. Furthermore, the landing velocity could be reduced to near zero with the ballast dropped at the designed ballast drop altitude. However, for the situations that did not use the control strategies, the trajectory of the balloon system could not be controlled from the ground after launch. This can be a cause of potential accidents.

2. The ballast flow rate mainly depended on the ballast drop altitude for each flight mission. The higher the ballast drop altitude, the smaller the ballast flow, which can help to design a ballast device. However, the higher the ballast drop altitude, the bigger the horizontal motion range, making recovery relatively more difficult.

These results can help with the overall design of a balloon flight system and planning a flight trajectory in advance. With the control strategies applied, this kind of flight mission can stay within the limited airspace, assisting the balloon system's rapid and safe recovery.

Table 2 shows the specifications of the missions. Figures 16–19 show the balloon system's change in vertical speed vs. time and change in altitude vs. time while the control strategies were applied (or not). Figure 20 shows the trajectories of the different missions.

**Table 2.** Flight mission specifications.

| Flight Mission | Floating Time (min) | Ballast Drop Altitude (km) | Ballast Flow Rate (kg·s$^{-1}$) |
|---|---|---|---|
| Mission A | 30 | 10 | 1.023 |
| Mission B | 30 | 8 | 1.671 |
| Mission C | 60 | 10 | 1.024 |
| Mission D | 60 | 8 | 1.672 |

The launch took place at latitude 39° and longitude 105° on August 8, 2019 (UTC + 16). The initial helium mass was 626 kg.

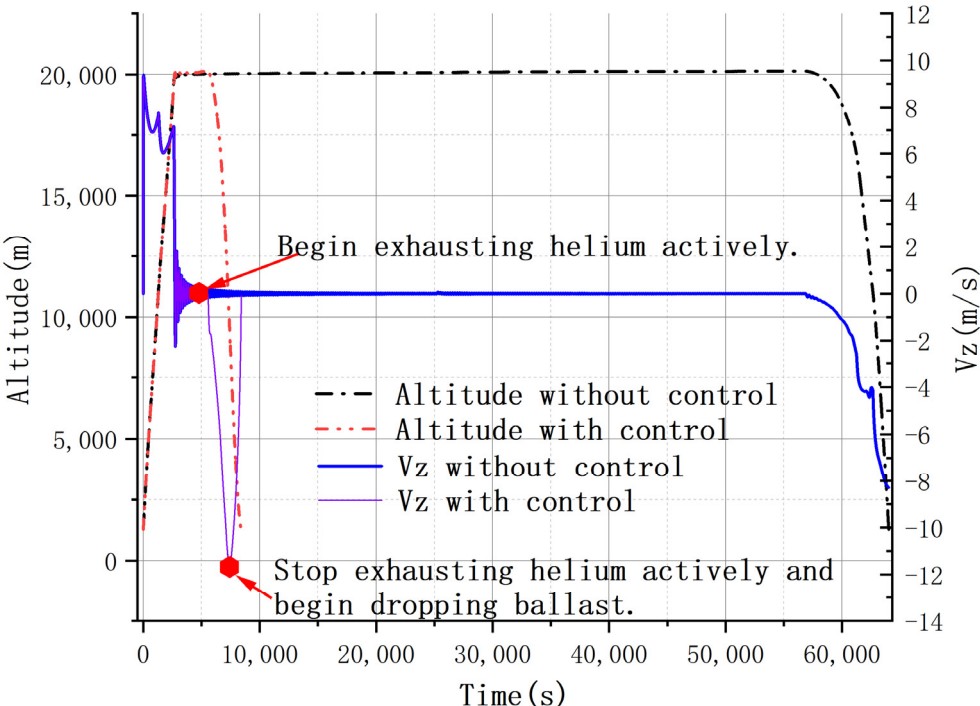

**Figure 16. Simulation results of Mission A:** balloon system's change in vertical speed vs. time and change in altitude vs. time while the control strategies are applied (or not).

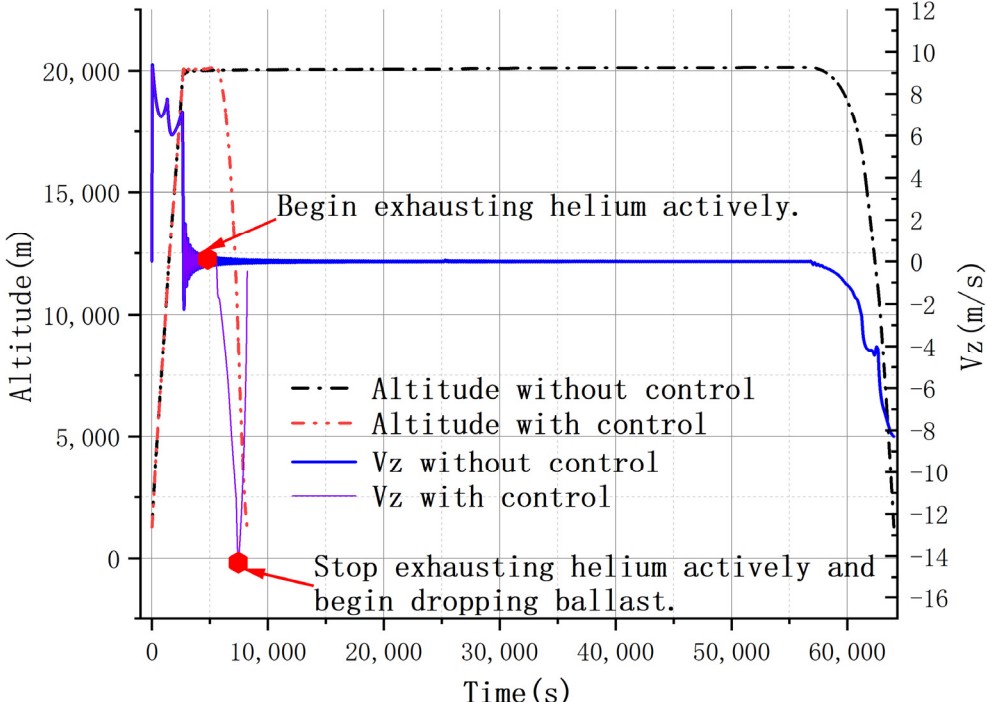

**Figure 17. Simulation results of Mission B:** balloon system's change in vertical speed vs. time and change in altitude vs. time while the control strategies are applied (or not).

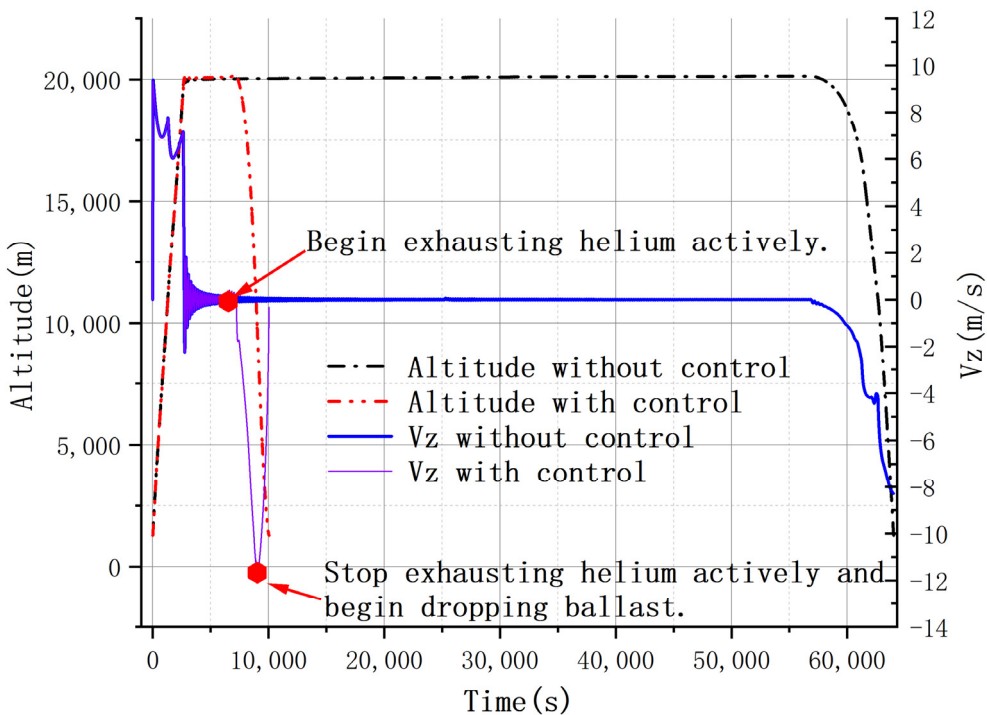

**Figure 18. Simulation results of Mission C:** balloon system's change in vertical speed vs. time and change in altitude vs. time while the control strategies are applied (or not).

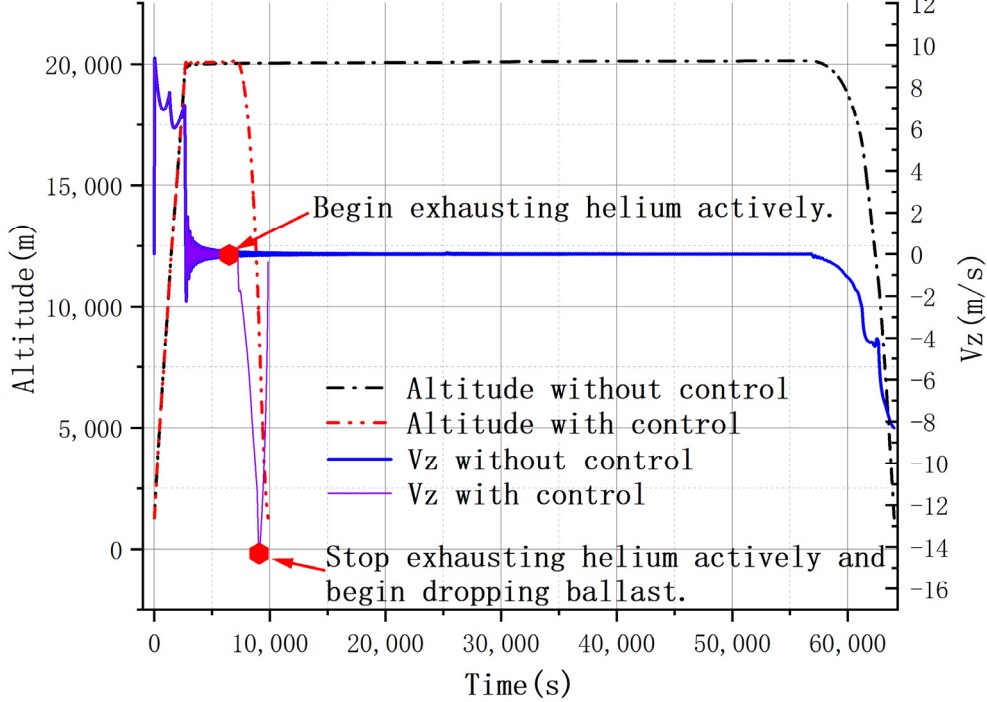

**Figure 19. Simulation results of Mission D:** balloon system's change in vertical speed vs. time and change in altitude vs. time while the control strategies are applied (or not).

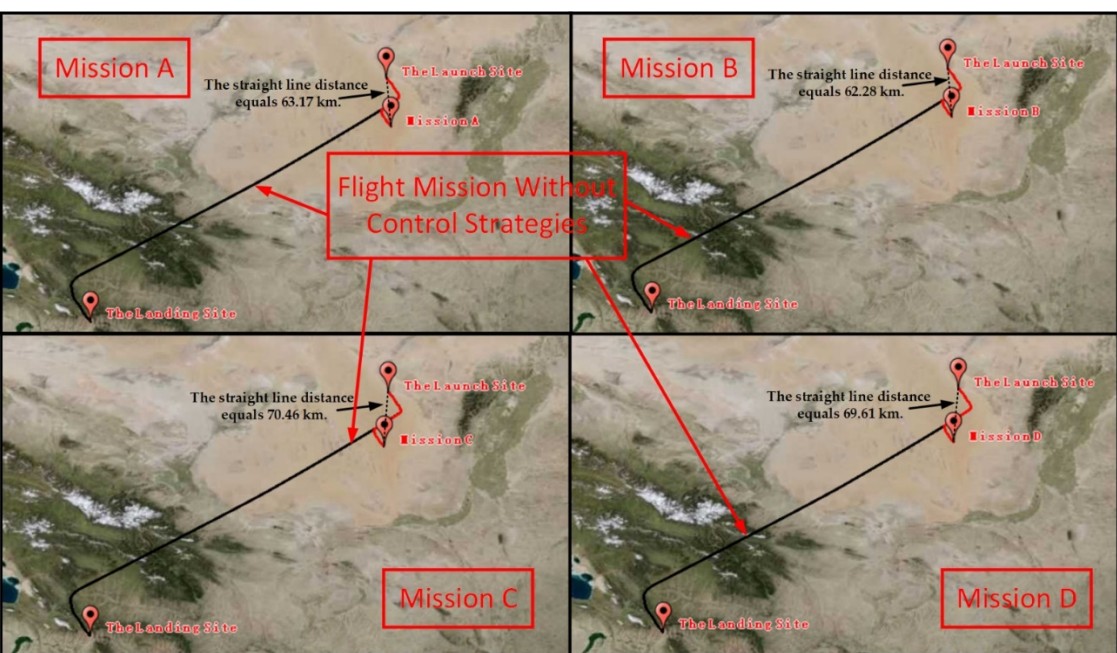

**Figure 20.** Trajectory of different flight missions discussed in the text.

## 5. Conclusions

A high-altitude zero-pressure balloon flight system flies with the wind in a horizontal direction, so it is necessary to predict its trajectory in advance to prevent the system from flying out of the required airspace and to avoid possible accidents. Most balloon flight systems need to fly within a limited airspace and accomplish rapid recovery to save on flight mission costs.

In the present work, a high-altitude zero-pressure balloon flight system with no parachute was simulated to predict trajectory, determine the impact of changing various parameters on balloon flight performance, and assist with rapid recovery. In order to further verify its accuracy, the simulation model needs to be validated by flight test data in the following research. The results can help in the planning of flight missions as well as design improvement for original equipment manufacturers of balloon flight systems.

The conclusions of the work are as follows:

1. During the ascent phase, the vertical velocity takes on a W shape. By the time the balloon flight system reaches the designed float altitude, the vertical velocity decreases sharply to 0 m/s and oscillates around 0 m/s until it finally drops gradually to 0 m/s.
2. During the ascent phase, the phenomenon of supercooling occurs, whereas the phenomenon of superheating occurs during the floating and descending phases. This information can help with balloon structure design.
3. The launch time plays an important role in trajectory prediction and has a significant impact on flight performance. The results show that the earlier the launch time, the longer the flight time and flight distance. However, the maximum flight time is less than 24 h. Furthermore, the launch time has no effect on the floating altitude and landing vertical velocity.
4. Maintaining the same launch time, changing the initial mass of helium has almost no effect on trajectory prediction. However, more helium filled on the ground will cause excessive vertical speed when ascending, leading to severe supercooling.
5. Based on realistic flight mission requirements, the floating mission time, horizontal motion range, and landing speed must be taken into consideration. The exhaust helium valve and ballast drop device can be used together to plan the trajectory. The simulation results show that with control strategies applied, flight missions can be

completed within limited airspace, and flight trajectories can be predicted to assist rapid and safe recovery and prevent harm from potential accidents.

6.  The ballast flow rate can be determined by the ballast drop altitude for each flight mission by experimentally dropping the ballast device before the balloon flight mission.

**Author Contributions:** Conceptualization, J.T. and S.P.; methodology, S.P. and J.T.; software, S.P.; validation, J.T.; formal analysis, S.P.; investigation, P.Y. and W.X.; resources, S.P. and J.T.; data curation, Y.L.; writing—original draft preparation, P.Y. and B.H.; writing—review and editing, S.P. and J.T.; supervision, J.T.; project administration, J.T.; and funding acquisition, J.T. All authors have read and agreed to the published version of the manuscript.

**Funding:** This work is supported by the National Natural Science Foundation of China under grant number 51906141 and the Shanghai Sailing Program under grant number 19YF1421500.

**Data Availability Statement:** Not applicable.

**Conflicts of Interest:** The authors declare no conflict of interest.

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
