# Peer review of "Research on Trajectory Prediction of a High-Altitude Zero-Pressure Balloon System to Assist Rapid Recovery"

_aerospace, doi:10.3390/aerospace9100622_

Round 1
Reviewer 1 Report
The paper presents a simulation model for the trajectory prediction of high-altitude zero-pressure balloon and a strategy to control this trajectory. The topic is interesting.
The paper presents in a sistematic way the modelling of the system, but improvements are needed. The literature review should be extended and some definition shall be provided to clearly specify the theoretical model. Above all, validation and conclusions shall be better substantiated and supported by results.
Detailed comments follow:
page 2, lines 48-59
The literature review only focuses on system modelling. The authors do not review the control strategies that are applied to this type of flight system.
page 4, line 131
The solar elevation angle “h” is defined and used in several equations. In principle, “h” could be negative (during the night). Are equations (6) – (7) – (15) – (16) valid also when h<0? I don’t think so, please clarify when the equations are valid or specify what happens during night time.
page 5, line 137
Is the “day serial number” computed starting from January 1st each year? It is not clear, please provide a definition of the “day serial number” computation.
page 5, lines 143-144
Please define “regional standard time” and “regional standard time position longitude”.
pages 5-6
Is there any references for equations from (9) to (19) and from (21) to (24)? If available, please mention them.
page 6, equation (24)
In equation (4) the direct solar radiation is denoted as Q_direct; in equation (24) as Q_sun. Please use the same notation throughout the paper
page 7, equation (29)
? has been already used in equation (20) to denote the average infrared emissivity factor of film material. Please use different symbols for first eccentricity and average infrared emissivity factor of film material.
page 7, lines 215-216
Assumption a) states “The horizontal components of the balloon`s speed and its speed relative to that of atmosphere winds are considered to be small”, whereas at page 9, lines 278-279 it is stated “For the horizontal motion, the balloon is regarded as floating with the wind, assuming that the horizontal velocity is equal to the wind speed magnitude”. I think that assumption a) should be “The horizontal components of the balloon`s speed are equal to the horizontal components of the wind speed”.
page 8, equations (31) and (32)
In equations (1) to (3) the altitude is denoted with “z”; in equations (31) and (32) the altitude is denoted with “x”. Please use the same symbol to denote the altitude throughout the paper.
page 11, Figure 5
It seems that the inputs to the block “Dynamic Model” do not include the atmospheric parameters that are required to compute the virtual mass, neither the virtual mass itself. Where is this parameter computed and using which inputs? Please clarify
page 11-12, section 4.1
The validation of the model is only based on qualitative consideration and some graphics. I suggest to provide also some metrics (for example rms or integral of the difference along the trajectory) to evaluate the performance of the proposed simulation model with respect to the previous model.
It would be great if the performance are also assessed against flight data, if available in the literature (as it was done in reference 27)
page 17, lines 475-481
According to me the presented conclusions are not clearly explained and substantiated. The flight missions are performed in different days and at different times of the day. Which parameter does affect the performance of the flight, the launch date ore the hour of the launch? In the conclusions it seems the first one is more relevant, in my view it should be the second one (if atmospheric conditions change in the same way during the day, independently from the date). Could you please elaborate a little more about these conclusions?
page 19-20, section 4.3
For each mission I suggest to compare the trajectory obtained when the control strategy is applied with the one obtained without control strategy (or with a standard control strategy, if available in the literature). It allows to assess the performance of the proposed strategy. For the sake of clearness, it would be useful to have the mission profile (for example altitude versus time) with the control strategy events highlighted (start enforcing the active exhaust procedure, start releasing the ballast)
page 22, lines 569-572
This conclusion shall be better elaborated (see comment to lines 475-481)
page 22, lines 573-578
This conclusion shall be better substantiated, by showing that the use of the proposed control strategy effectively introduces an improvement with respect to the mission performed without a control strategy. See also comment to section 4.3.
Reviewer 2 Report
I've attached some written edits for you to consider.

Round 2
Reviewer 1 Report
Dear Authors, thank you for answering all my comments.
I have some further suggestions to improve the paper:
1) The added figures 16 to 19 allow highlighting the effect of the control strategy that shorten the mission duration. However, since the descending phase of the mission without control strategy is not shown, it is not possible to compare the descending phase with and without control. I suggest to show in the figure also the descending phase without control. You can do it by extending the time range or by showing in different figures the comparison of the most relevant mission phases.
2) If I correctly understand, the added figure 20 shows four different missions, all performed using the control strategy. I tink that it is more relevant if for each mission you show the comparison with and without the application of the control strategy. I suggest to use a different figure for each mission comparison, otherwise the figure is not readible
3) I understand that actual flight data are not available in the literature. However I think that a validation based on experimental data is a mandatory step for each simulation model. I suggest to state in the model validation section and in the conclusion that the validation of the model is preliminar and that the future step of the research will be the validation of the model through experimental data
Reviewer 2 Report
Line 616 - change "effects" to "effect". No other edits to suggest.
